# Automatic Assist Level Adjustment Function of a Gait Exercise Rehabilitation Robot with Functional Electrical Stimulation for Spinal Cord Injury: Insights from Clinical Trials

**DOI:** 10.3390/biomimetics9100621

**Published:** 2024-10-13

**Authors:** Ryota Kimura, Takahiro Sato, Yuji Kasukawa, Daisuke Kudo, Takehiro Iwami, Naohisa Miyakoshi

**Affiliations:** 1Department of Orthopedic Surgery, Akita University Graduate School of Medicine, Akita 010-8543, Japan; twrofwsh0930@outlook.com (T.S.); miyakosh@doc.med.akita-u.ac.jp (N.M.); 2Department of Rehabilitation, Akita University Hospital, Akita 010-8543, Japan; kasukawa@doc.med.akita-u.ac.jp (Y.K.); dkudo@doc.med.akita-u.ac.jp (D.K.); 3Department of Systems Design Engineering, Faculty of Engineering Science, Akita University Graduate School of Engineering Science, Akita 010-8502, Japan; iwami@gipc.akita-u.ac.jp

**Keywords:** spinal cord injury, functional electrical stimulation, rehabilitation robot, reinforcement learning, assistive level

## Abstract

This study aimed to identify whether the combined use of functional electrical stimulation (FES) reduces the motor torque of a gait exercise rehabilitation robot in spinal cord injury (SCI) and to verify the effectiveness of the developed automatic assist level adjustment in people with paraplegia. Acute and chronic SCI patients (1 case each) performed 10 min of gait exercises with and without FES using a rehabilitation robot. Reinforcement learning was used to adjust the assist level automatically. The maximum torque values and assist levels for each of the ten walking cycles when walking became steady were averaged and compared with and without FES. The motor’s output torque and the assist level were measured as outcomes. The assist level adjustment allowed both the motor torque and assist level to decrease gradually to a steady state. The motor torque and the assist levels were significantly lower with the FES than without the FES under steady conditions in both cases. No adverse events were reported. The combined use of FES attenuated the motor torque of a gait exercise rehabilitation robot for SCI. Automatic assistive level adjustment is also useful for spinal cord injuries.

## 1. Introduction

Worldwide, approximately 15.4 million people have been living with spinal cord injury (SCI). SCI is a major cause of long-term disability, accounting for over 4.5 million years of life lived with disability in 2021 [1]. Many of these injuries are reported to result from high-energy trauma, such as motor vehicle accidents and falls. In contrast, the number of cases of incomplete SCI due to low-energy trauma from falls on level surfaces in older adults is increasing in Japan [2]. In particular, in the case of incomplete paralysis, appropriate assistance is needed in rehabilitation, in accordance with motor learning theory, depending on the recovery process of motor paralysis. A basic premise of motor learning is that of high-volume repetition and task-oriented training. Treatments based on this premise have become a major focus of research on the recovery of motor function in central nervous system disorders such as spinal cord injury and stroke [3,4]. For high-frequency training, robotic rehabilitation can provide more extensive training while compensating for the patient’s lost functions and reducing the burden on supporting medical staff.

Neuromodulation techniques, such as functional electrical stimulation (FES), are important tools in restorative neurology [5]. FES can noninvasively activate paralyzed muscles or muscle groups through electrodes placed on the skin. Depending on whether the upper or lower motor neurons are damaged, stimulation directly activates the motor nerves or muscle fibers. Nerve stimulation relies on intact peripheral nerves and neural signal processing in the intact portion of the spinal cord below the lesion [6]. Kralj et al. [7] developed an FES method and system to facilitate ambulation in individuals with spinal cord injury. Furthermore, FES has been used in combination with ankle–foot orthosis [8] and hip–knee–ankle–foot orthosis [9]. There have also been several reports of the addition of FES into gait training rehabilitation robots [10,11], and it has been reported to reduce the electric motor torque of an exoskeletal assistive walking robot [12]. A systematic review and network meta-analysis has also indicated that FES was the most effective treatment for improving walking velocity and distance in incomplete spinal cord injury [13]. Furthermore, combining FES with a robot resulted in exercise with less muscle fatigue than FES alone [14], making hybrid FES–robot training a viable option for prolonged exercise. As a result, the use of a gait rehabilitation robot with FES improves the range of motion of the joints, the muscle strength and the ability to walk [15,16]. Robot-assisted training with FES appears to support the recovery of residual function after SCI and has been observed to lead to improvements in motor function and strength in the lower extremities [17]. Previously, we developed a gait training rehabilitation robot with FES [18] and confirmed that the robot’s torque was reduced by using FES in pseudo paraplegics [19]. In addition, a function for automatically adjusting the level of assistance using reinforcement learning has been developed, and its effectiveness in healthy subjects has been confirmed [20].

It is unknown whether using FES in combination with a gait training rehabilitation robot reduces motor torque in people with paraplegia with spinal cord injury. Moreover, there are currently no devices that automatically adjust the level of assistance. This study aimed to verify whether concurrent use of FES reduces the motor torque of a gait exercise rehabilitation robot in SCI and the effectiveness of the developed automatic assist level adjustment in people with paraplegia.

## 2. Materials and Methods

Patients with paraparesis were recruited from our institution. Paraparesis was defined as those with SCI because of trauma. The inclusion criteria were as follows: (1) inability to walk unaided; (2) recognition of the significance of this research and participation in the research of their own free will; and (3) written informed consent after explanation in the informed consent document. The exclusion criteria were as follows: (1) inability to follow the therapist’s instructions, (2) conditions in which exercise load leads to deterioration of physical condition, (3) severe joint contractures or deformities, and (4) other movement limitations for any reason.

Patients performed 10 min of gait exercises with and without FES using a rehabilitation robot. The robot’s exoskeleton was designed based on the hip–knee–ankle–foot orthosis for paraplegia. The trunk, thigh, and lower leg were secured with belts. An ankle-foot orthosis (RAPS, Tomei Brace, Seto, Japan) was used for the ankle joint, allowing for angle adjustment based on the patient’s spasticity. The actuators driving the hip and knee joints were equipped with encoders, enabling the acquisition of joint angle data. Although the entire exoskeleton weighed 40 kg, it did not affect the patient during use owing to the weight being offset by a counterweight. The orthosis could be adjusted in length to fit each patient’s thigh and lower leg. The patients were lifted, and their weight was supported by a rehabilitation lift (SP-1000, Moritoh, Ichinomiya, Japan); following this, they were walked under this condition on a treadmill (8.1T, JohnsonHealth Tech Japan, Tokyo, Japan). Bilateral quadriceps and hamstrings were stimulated using FES (Dynamid, DM2500, Minato Medical Science, Osaka, Japan) (Figure 1). The quadriceps, primarily the rectus femoris, were stimulated from mid-swing to mid-stance during the robotic gait cycle. The stimulation point was located at the motor point, identified by palpation of the anterior superior iliac spine and lateral femoral condyle. The hamstrings, mainly on the lateral side, were stimulated from pre-swing to mid-stance, and the stimulation point was located at the motor point identified by palpation of the sciatic tuberosity and the head of the fibula. The exoskeleton system was pre-programmed using gait data from the joint angles of a healthy individual. The system performs walking motions by changing the positions of the hip and knee joints according to the gait data. The stimulus intensity was set to the lowest stimulus (15–20 mA, 25 Hz) that produced joint movement, and the stimulus timing was synchronized to the gait cycle [19].

In this device, the motor drive was controlled by a computer to reproduce the walking motion. The motor’s output torque (Nm) was proportional to the stiffness parameter, and the range was divided into 50 parts, defined as the assist level. The higher the assistance level, the greater the amount of assistance. Force control was used to control the motor, and compliance control was used to vary the amount of assistance. The control Equation (1) used for compliance control is shown below.
(1)$$\tau =K\cdot \theta +C\cdot \dot{\theta }+I\cdot \overset{..}{\theta }$$

In the equation, *K* (Nm/deg) represents the stiffness; *C* (Nm/deg∙s) represents the viscosity; *I* (Nm/deg ∙ s^2^) represents the inertia; *θ* (deg) represents the deviation between the target joint angle and measured angle; $\dot{\theta }$ (deg/s) and $\overset{..}{\theta }$ (deg/s^2^) represent first and second derivative values of *θ*, respectively; and *τ* (Nm) is the motor torque. The equation follows the equation of motion, and the first term indicates that the motor output increases as the difference between the target joint angle and measured joint angle increases. In the first term, the output torque is proportional to the stiffness parameter *K*, i.e., when the stiffness parameter represented by *K* is large, the output torque increases; conversely, when *K* is small, the output torque decreases. In this study, the ranges of stiffness, viscosity, and inertia parameters were determined experimentally based on a previous study [16] and the range of stiffness (*K*) was divided into 50 parts and defined as the assist level. The assistance level provided feedback to the patient through data displayed on the computer monitor placed in front of the robot.

Reinforcement learning was used to adjust the assist level automatically. The method used was Q-learning, a type of off-policy temporal difference learning, and the *ε*-greedy method was used to determine the policy [21]. In this study, the reinforcement learning environment was defined as “the device and the entire subject wearing it”. The subject was trained to select actions by choosing between three options: increasing, maintaining, or decreasing the level of assistance. The subject was rewarded with angular error and motor torque for each walking cycle. The *ε*-greedy method introduced a search rate *ε* (0 ≤ *ε* ≤ 1) into the decision process and prevented the system from falling into a local solution through exploration by selecting a random action with probability *ε* regardless of the action value and an action corresponding to the maximum action value with probability 1 − *ε*. In the initial stages of learning, we increased the proportion of exploration by setting *ε* to a large value and collecting knowledge. Then, in the advanced stages of learning, we set *ε* to a small value so that the robot can select the optimal action by using the collected knowledge. The Q-learning algorithm is an off-policy type and the aim was not to optimize the policy, but to optimize the action state value function Q, which indicates the effectiveness of actions, and to construct a decision-making standard that can select the optimal action for the environment. The update equation for updating the Q value *Q*(*s*_*t*_, *a*_*t*_) when action *a*_*t*_ is selected in state *s*_*t*_ is shown in Equation (2).
*Q*(*s_t_*, *a_t_*) ← *Q*(*s_t_*, *a_t_*) + *α*(*γ* + *γ* · *maxQ*(*s*_*t*+1_, *a*_*t*+1_) − *Q*(*s_t_*, *a_t_*))(2)

In Equation (2), *s* represents the state, *a* represents the action selected in state *s*, *α*(0 ≤ *α* ≤ 1) represents the learning coefficient, *r* represents the reward obtained as a result of the action, *γ*(0 ≤ *γ* ≤ 1) represents the discount rate, and *maxQ*(*s*_*t* + 1_, *a*_*t* + 1_) represents the maximum Q value of the actions that can be selected in the next action. The action value function Q for selecting action *a* in a given state *s* is expressed as *Q*(*s*, *a*). The Equation (2) means that if the value of the reward r, obtained as a result of the action, is positive, then the value of Q(*s*, *a*) is increased, and if the reward is negative, the value of Q(*s*, *a*) is decreased. By repeating the process of taking actions and updating the Q value using the Equation (2), it becomes possible to proceed with learning about the Q value. In addition, in the Equation (2), the state *s*, action *a*, and reward *r* represent variables that are brought about by interaction with the environment, so there was no need to adjust the parameters. The parameters that had be adjusted here were the learning coefficient *α* and the discount rate *γ*. The actions that the agent could take were classified as increasing, maintaining, or decreasing the assist level, and the corresponding Q values were set for each. Because the *ε*-greedy method was used here to determine the policy, the action corresponding to the largest Q value among each of these was selected, while a random action was selected with probability *ε*, and the Q value was updated according to the reward obtained from the environment as a result. By repeating this process, the Q values were updated to minimize the reward *r* obtained, thereby reducing the discrepancy between the joint angle and motor torque during the walking cycle. This allowed the assist level to be adjusted to an optimal level for the patient.

The walking speed was set at 0.8 km/h. The maximum torque values and assist levels for each of the 10 walking cycles when walking became a steady condition were averaged and compared with and without FES. All statistical analyses were conducted using EZR (Saitama Medical Center, Jichi Medical University, Saitama, Japan) [22]. The motor torque and assist level were compared using the paired *t*-test, with statistical significance set at *p* < 0.05.

This study was approved by our institution’s ethics committee (approval number: CRB2180005). All of the individuals voluntarily participated in the study and provided written informed consent.

## 3. Results

The subjects were a patient with acute thoracic spinal cord injury (33-year-old man), 2 weeks after injury, American Spinal Injury Association (ASIA) Impairment Scale (AIS) C, neurological level of injury (NLI) T12 without spasticity, and a patient with chronic thoracic spinal cord injury (34-year-old man), 2 years after injury, AIS C, and NLI T11 with spasticity (modified Ashworth scale: grade 2).

Assist level adjustment allowed both the motor torque and assist level to decrease gradually to a steady state. Each value reached a steady state between 60 and 120 s. The motor torque was significantly lower with the FES than without the FES under steady conditions in both cases (Table 1). Furthermore, the assist levels were significantly lower with FES than without FES in both cases (Table 2).

The electrical stimulation delivered via the FES did not cause any adverse effects, such as pain, and did not lead to any adverse events associated with robotic gait exercises.

## 4. Discussion

The combined use of FES attenuated the motor torque of the gait exercise rehabilitation robot for spinal cord injury. Furthermore, automatic adjustment of the assistance level using reinforcement learning proved to be effective in gait exercises for patients with spinal cord injuries, and the combined use of FES attenuated the assistance level. It was shown that the automatic assist level adjustment system could be used in conjunction with the torque generated by the FES. This suggests that the intrinsic muscle activity generated by FES reconstructed some of the torque required for walking (Figure 2). In conventional robotic gait training, the gait is reconstructed by combining the robot torque with the patient’s muscle torque; the use of FES in conjunction may help attenuate the robot torque.

In functional electrical stimulation therapy (FEST), three factors are crucial: the patient, FES, and the therapist [23]. A phase I randomized control trial (RCT) has revealed that locomotion function improved significantly more with FEST than a non-FEST controlled intervention [24]. The therapist could be replaced by a robot. Regarding the combination of FES and gait rehabilitation robots, we have reported that the smoothness of movement was not lost even when FES was used in combination [18], and that the robot torque was reduced by pseudo-paraplegia [19]. The present study builds on these previous reports and extends them by demonstrating that the combined use of a gait rehabilitation robot and FES reduces the motor torque of the robot in SCI. In order to substantiate the clinical efficacy of FEST when utilizing robots, it is necessary to conduct RCTs; for example, to compare FEST with FES monotherapy. Furthermore, the potential of combining robotics and FES in rehabilitating patients with disorders affecting the central nervous system remains to be fully validated. Future validation is needed because robotic rehabilitation is expected to be integrated with brain–computer interface (BCI) [24] in the future.

Although machine learning and reinforcement learning in exoskeletal rehabilitation robots have previously been studied and reported on [25,26,27], this is the first report of their combination with FES for SCI. The automatic and appropriate adjustment of task difficulty according to the degree of paralysis is an effective rehabilitation tool from the perspective of motor learning [28]. Furthermore, the patients did not experience any discomfort during gait, and no adverse events were observed. Further research is required to ascertain the clinical efficacy of this system’s rehabilitative intervention.

The integration of FES into the robot resulted in a reduction in the torque of the motor. This could ultimately result in a reduction in power consumption and size. The large size of conventional gait rehabilitation robots has been a substantial barrier to their implementation in a broader range of settings. It is of paramount importance to reduce the size of the robot if the objective of achieving the generalization of robotic rehabilitation is to be met. The incorporation of artificial muscles may prove an efficacious solution to these challenges [29]. Given that the concurrent use of an exoskeleton-type robot and a treadmill represents a safe method for gait training rehabilitation, it is imperative to reduce the size of each to promote the overall effectiveness of robotic rehabilitation. As FES reduces the torque generated by the robot’s motor, it seems reasonable to employ FES to reduce the overall size of the robot. Consequently, the treadmill size can be reduced, thereby increasing compatibility with the treadmills that are commonly used. This may, in turn, facilitate the implementation of gait training rehabilitation robots in a wide range of applications, including the prevention of disabilities in the aging population. From this study, clinicians should consider combining FES with a gait rehabilitation robot, including the clinical efficacy of FES in SCI.

This study has several limitations. First, it included only two cases, but data were available for both acute and chronic cases. The automatic assistance level adjustment system proved useful in both cases. However, further validation of the clinical effects of rehabilitation with a sufficient number of patients is needed. Second, the walking speed and FES settings were fixed; further verification of the variations caused by changes in speed and FES settings is needed.

## 5. Conclusions

The combined use of a gait rehabilitation robot and FES reduced the robot’s motor torque in SCI, and automatic assist level adjustment through reinforcement learning was effective in people with paraplegia. The clinical outcomes need to be evaluated.

## Figures and Tables

**Figure 1 biomimetics-09-00621-f001:**
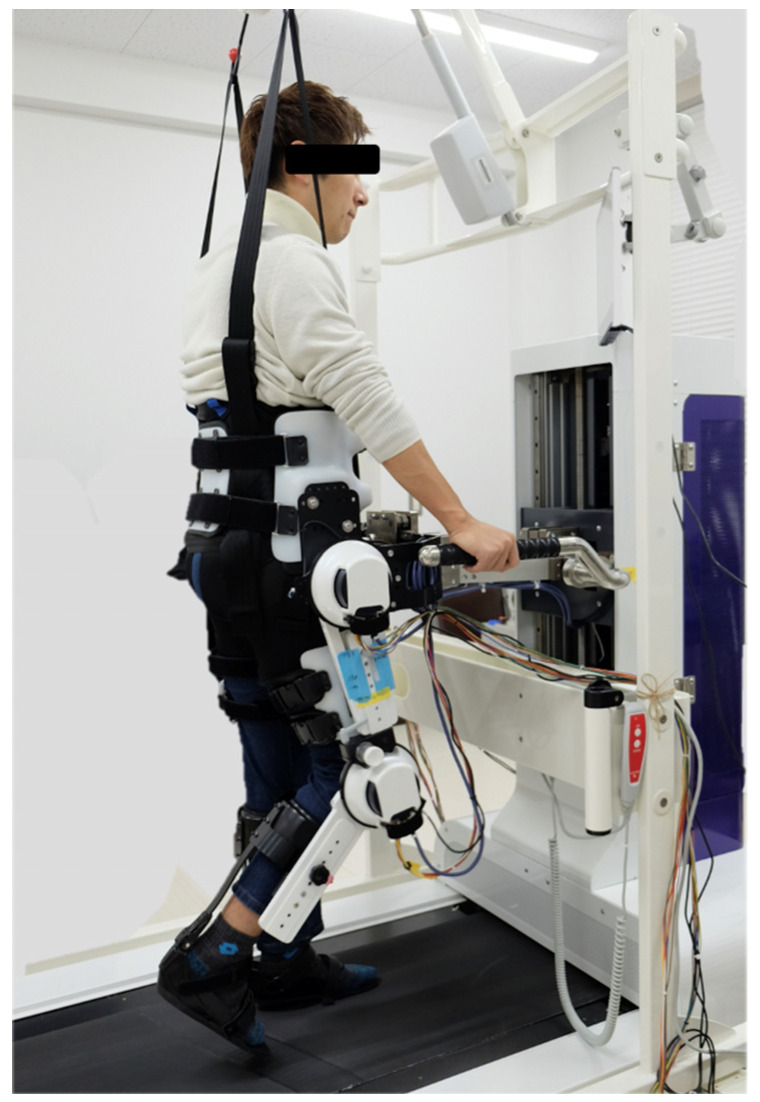
Gait exercise rehabilitation robot. The robot has an exoskeleton, rehabilitation lift, treadmill, and functional electrical stimulation (FES).

**Figure 2 biomimetics-09-00621-f002:**
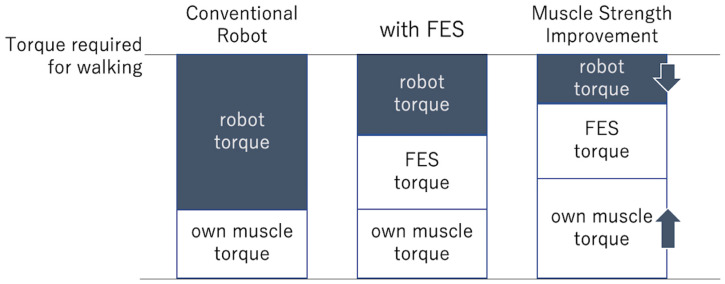
The motor torque. Conventional robots were constructed using the robot’s torque and their own muscle’s torque in order to prove the torque required for walking. Motor torque is attenuated by the combined use of functional electrical stimulation (FES). Furthermore, the required motor torque decreases as the patient’s muscle torque improves.

**Table 1 biomimetics-09-00621-t001:** Motor torque.

(Nm)	Case 1			Case 2		
	FES (−)	FES (+)	*p*	FES (−)	FES (+)	*p*
Hip Right	18.6 ± 1.5	16.6 ± 1.7	0.0237	17.9 ± 1.2	8.7 ± 1.2	<0.001
Hip Left	18.1 ± 1.4	16 ± 1.6	0.0181	14.7 ± 1.3	11.9 ± 1.7	<0.001
Knee Right	20.3 ± 1.8	13.1 ± 1.5	<0.001	15.4 ± 1.2	13.4 ± 1.6	0.0047
Knee Left	19.2 ± 1.4	17.1 ± 1.9	0.0226	18.4 ± 1.5	12.2 ± 1.3	<0.001

**Table 2 biomimetics-09-00621-t002:** Assist levels.

(Nm)	Case 1			Case 2		
	FES (−)	FES (+)	*p*	FES (−)	FES (+)	*p*
Hip Right	28.5 ± 0.5	26.5 ± 0.5	<0.001	42.9 ± 0.3	30.9 ± 0.7	<0.001
Hip Left	24.1 ± 0.6	23.4 ± 0.7	0.0445	38.9 ± 0.3	30.9 ± 0.3	<0.001
Knee Right	30.8 ± 0.4	23.9 ± 0.3	<0.001	25.4 ± 0.5	20.6 ± 0.5	<0.001
Knee Left	29.5 ± 0.5	17.5 ± 0.5	<0.001	33.9 ± 0.3	21.4 ± 0.5	<0.001

## Data Availability

Data are contained within the article. Contact the author for additional data.

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
