# Peer review of "Automatic Assist Level Adjustment Function of a Gait Exercise Rehabilitation Robot with Functional Electrical Stimulation for Spinal Cord Injury: Insights from Clinical Trials"

_biomimetics, 2024, doi:10.3390/biomimetics9100621_

Round 1
Reviewer 1 Report
Comments and Suggestions for Authors
1. It would be helpful to include more details about the selection criteria for acute and chronic SCI patients.
2. The methodology for FES and robotic assistance is well-explained. Consider providing more information on the reinforcement learning algorithm for assist level adjustment.
3. The statistical analysis is not clear. Need to explain the data normalizing and statistical test.
4. Consider providing more information on the reinforcement learning algorithm for assist level adjustment.
5. it would be beneficial to elaborate on the clinical implications of the findings and how they could influence rehabilitation practices.
6. What is a home message?
7. Write the discussion based on the SWOT.
8. Write the limitations and strengths.
Author Response
We would like to thank the reviewer for your constructive critique to improve the manuscript. We have made every effort to address the issues raised and to respond to all comments. The revisions are indicated in red font in the revised manuscript. Please, find below a detailed, point-by-point response to the reviewer's comments. In addition, we made revisions and additions to ensure that the manuscript was at least 4,000 words or more and included at least 30 references, in line with the guidelines.
Reviewer 1
- It would be helpful to include more details about the selection criteria for acute and chronic SCI patients.
We added information about the inclusion and exclusion criteria in Materials and Methods. We also included the patient details in the results section.
- The methodology for FES and robotic assistance is well-explained. Consider providing more information on the reinforcement learning algorithm for assist level adjustment.
We added information on the reinforcement learning algorithm for assist level adjustment. (Lines 132-170)
- The statistical analysis is not clear. Need to explain the data normalizing and statistical test.
Lines 171 to 176 describe the handling of data and statistical analysis.
- Consider providing more information on the reinforcement learning algorithm for assist level adjustment.
Same as Comment 2, we added information on the reinforcement learning algorithm for assist level adjustment. (Lines 132-170)
- it would be beneficial to elaborate on the clinical implications of the findings and how they could influence rehabilitation practices.
We added a description of the clinical implications of the survey results and their potential impact on rehabilitation practice. (Lines 236-237 and 249-251)
- What is a home message?
Our home message is “The combined use of FES attenuated the motor torque of a gait exercise rehabilitation robot for SCI”. We described it in Conclusions.
- Write the discussion based on the SWOT.
We edited the discussion to match the SWOT.
- Write the limitations and strengths.
We added a note about limitations and strengths to our discussion.
Reviewer 2 Report
Comments and Suggestions for Authors
The authors performed a clinical trial with two SCI patients using the developed gait exercise rehabilitation robot with and without FES. This work seems to be a clinical trial version of the previously published research, with automatic assist level adjustment functionality by reinforcement learning. The presentation style is clear and concise and the clinical trial seems successful, even though only two patients are tried. The following issues must be addressed to be justified for journal publication.
1. I suggest the authors include clinical trials in the title.
2. On page 3, there is no equation number, and theta_1, theta_2 must be changed to theta^dot, theta^ddot.
3. On page 3, there are brief statements regarding reinforcement learning implementation. The problem formulation must be defined, including governing equations, constraints, and objective functions with input values, with appropriate justification for the automatic torque level adjustment. Also, the simulation process and data for optimal policy must be clarified.
4. In the discussion, lines 144-170 must be included in the introduction for the reader to distinguish the work from the previous literature.
Author Response
We would like to thank the reviewer for your constructive critique to improve the manuscript. We have made every effort to address the issues raised and to respond to all comments. The revisions are indicated in red font in the revised manuscript. Please, find below a detailed, point-by-point response to the reviewer's comments. In addition, we made revisions and additions to ensure that the manuscript was at least 4,000 words or more and included at least 30 references, in line with the guidelines.
Reviewer 2
- I suggest the authors include clinical trials in the title.
I changed the title to “Automatic Assist Level Adjustment Function of a Gait Exercise Rehabilitation Robot with Functional Electrical Stimulation for Spinal Cord Injury: Insights from Clinical Trials”
- On page 3, there is no equation number, and theta_1, theta_2 must be changed to theta^dot, theta^ddot.
I added the equation number and changed the equation to the following.
? = ? ∙ ? + ? ∙ + ? ∙
- On page 3, there are brief statements regarding reinforcement learning implementation. The problem formulation must be defined, including governing equations, constraints, and objective functions with input values, with appropriate justification for the automatic torque level adjustment. Also, the simulation process and data for optimal policy must be clarified.
We made additions and corrections to the reinforcement learning section. (Lines 132-170)
- In the discussion, lines 144-170 must be included in the introduction for the reader
to distinguish the work from the previous literature.
As you pointed out, we revised the introduction and discussion.
Round 2
Reviewer 1 Report
Comments and Suggestions for Authors
The author responds to the comments.